# When Movement Moves: Study Protocol for a Multi-Method Pre/Post Evaluation Study of Two Programmes; the Danish Team Twin and Cycling Without Age

**DOI:** 10.3390/ijerph181910008

**Published:** 2021-09-23

**Authors:** Andreas Jørgensen, Christina Bjørk Petersen, Martin Eghøj, Mette Toftager

**Affiliations:** National Institute of Public Health, University of Southern Denmark, 1455 Copenhagen, Denmark; chrb@sdu.dk (C.B.P.); maeg@sdu.dk (M.E.); Mto@sdu.dk (M.T.)

**Keywords:** study protocol, multi-methods, disability evaluation, disabled persons, functionally-impaired elderly, healthy ageing exercise, quality of life, Cycling Without Age, Team Twin

## Abstract

Physical activity (PA) can improve physical, mental, and social health, leading to quality of life (QoL). However, some are unable to participate independently due to age-related impairments or disabilities. This study protocol presents the design, methods, outcomes, strengths and limitations of the study “When Movement Moves” (WMM). WMM investigates whether indirect PA in a social context, where persons are unable to participate independently, can result in outcomes similar to those of independent PA, by evaluating the effects of disabled’s, elderly, volunteers’, relatives’ and nursing staff’s participation in either the running programme (Team Twin) or cycling programme (Cycling Without Age). Both programmes seek to enhance QoL through indirect PA in a social context, making PA possible for elderly and disabled persons through PA conducted by abled-bodied volunteers. WMM is a multi-method 12–16-week pre/post evaluation with quality of life, physical, mental and social health as main outcomes. Pre/post measurements consist of questionnaires, clinical examinations, and physical and cognitive appraisal. Throughout, interviews and participant observations will be conducted. Combined results will provide essential knowledge on the effects and experiences of indirect PA. Explorative data will pave the way for further research. Findings may inform policies, guidelines and health promotion among the elderly and disabled.

## 1. Introduction

The elderly and individuals with functional disabilities often face multiple personal and environmental barriers reducing their ability to engage and participate in physical and social activities [1,2,3,4,5]. Increasing age and level of disability correlate with lower levels of physical activity (PA), and an increased prevalence of chronic and cardiovascular diseases [6,7,8,9,10,11], which can result in premature death [12,13,14,15]. Contemporary literature indicates a dose–response relationship between PA and promotion of all health outcomes: the greatest risk reduction comes with higher intensities of PA [16], thus, PA serves as disease prevention and health maintenance [6,17]. Moreover, the World Health Organisation encourages the elderly and disabled, regardless of age or impairments, to perform any kind of PA, as “some is better than nothing” [18]. Additionally, a recent review indicates that less PA than the recommended amount can enhance health outcomes among persons with a disability [19]. Moreover, a recent study has linked various alternative forms of PA and therapy to be beneficial for health promotion and rehabilitation among individuals with cerebral palsy (CP) [20]. The literature further indicates that PA can promote cognitive function among the elderly [21] and cognitively disabled [22] on top of the well-known physiological benefits of PA. Elderly and disabled persons participating in adapted sports experience an increase to mental and social health factors, e.g., decreased feelings of loneliness, increased feelings of self-efficacy [23], autonomy and well-being [24,25] and improvements in quality of life (QoL) [25,26,27,28,29]. Social support is identified as a crucial facilitator of the disabled and elderly’s participation in PA [30]. Prior studies, however, have mainly focused on individuals with spinal cord injuries or elderly who can participate in (elite) sport and PA by themselves, despite a disability [24,27,31] or physical limitations due to ageing [32]. There is a lack of literature on the frailest individuals in these groups, who are unable to participate in most kinds of PA. The few studies available have low external (i.e., generalisability) validity [27,33]. Pilot studies investigating indirect PA (defined as an activity where a person unable to partake in physical activity on their own, is joined by someone who is able, and together partake in PA) in the disabled [34] and elderly [35,36,37,38,39,40] populations exist, however, and these report promising effects of indirect PA on all health outcomes and QoL.

Despite the existence of pilot studies, little is known about the physical, mental and social health benefits of participation in indirect PA in a social context. Programmes (interventions/initiatives) providing indirect PA may motivate and promote participation if (some of) the same health benefits derived from PA can be attained through indirect PA, which may, in turn, result in improved QoL for the elderly and disabled.

Therefore, the research project “When Movement Moves” (WMM) evaluates the pre-existing Danish programmes: Team Twin (TT) and Cycling Without Age (CWA) as both programmes make use of volunteers to assist indirect PA in disabled individuals and the elderly, respectively, using equipment designed for the purpose. The programmes ultimately share the same goal: enhancement of QoL and promotion of physical, social and mental health achieved through indirect PA in a social context [41,42].

### 1.1. Presenting the Programmes

#### 1.1.1. Team Twin—Association and Members

TT is an overall umbrella association for the sports and inclusion concept “Team Twin—We run Together”. The vision of the association is to integrate disabled and able-bodied persons into a social running community. The first local TT club was established in 2014 in Copenhagen, Denmark. Today the association has nine local clubs across Denmark with approximately 500 members, as well as a club in Norway [41]. In the association, members are divided into three categories: Handiathletes (HAs) who are severely functionally limited persons, runners who are volunteers, and relatives, such as parents or siblings (of the athletes). The runners and HAs exercise together, with the athletes sat in chairs designed specifically for running, meaning the handiathletes participate (in)directly in activities where they are “moved by others”. The role of the relatives is to assist and support the athletes in getting ready and into the chair prior to the run, as well as helping them back into their wheelchair post-run. Persons with a self-reported disability are estimated to represent 30% of the younger Danish population (16–64 years), including impairments and long-term health conditions [8,43]. Of these, 10% report a severe physical disability. Conditions of everyday life (i.e., housing conditions, hours allocated for disabled carers, etc.) for persons with disabilities vary depending on the level of impairment (cognitive and physical), as does the need for assistance during everyday life [8]. The participating HAs have various disabilities, with CP as the most frequent. Common for all HAs is, despite different disabilities, that they are wheelchair dependent.

#### 1.1.2. Cycling Without Age—Association and Members

CWA is a concept and an association with a mission to make nursing home residents (re)achieve active citizenship. Their slogan is “Everyone has right to wind in their hair” [42]. CWA was invented in Copenhagen, Denmark, in 2012. Today it is a world-spanning association represented in more than 42 countries, with over 7.000 registered volunteers across 430 locations in Denmark, the majority of whom are associated with nursing homes. In 2018 alone, a total of 14.877 CWA trips were registered. In CWA, nursing home residents referred to as passengers, who have a low level of mobility volunteers, are taken on trips in rickshaw bikes by volunteers, referred to as pilots, on trips varying from spontaneous, short (less than an hour) trips to well-planned long (day-spanning) trips. Nursing home staff are part of the concept, too. Their role is to help the residents get ready for the trips [42]. Nursing home residents, such as the disabled younger population in TT, face physical, personal, and environmental barriers for PA. They often report lower well-being and a greater likelihood of experiencing social isolation [7,8,44,45,46] due to ageing or diminishing health conditions. Among the elderly, functional limitations and cognitive impairment are common [7]. Due to demographic changes, the proportion of elderly and physically limited elderly persons in the world is increasing [7]. Roughly 3.6% (41.000) of Danish citizens aged 65 or older live in nursing homes. Half of this population have chronic diseases, while 66% suffer from dementia [47].

These programmes may lead to improvements in QoL in the disabled and elderly, who are otherwise severely limited in their ability to partake in both physical activity and society.

#### 1.1.3. Interrelated Dependency

In both TT and CWA, an interrelated dependency between each programme’s three groups exist (Figure 1). The activities cannot function as intended if one or more groups are not present. For example, the runners cannot further the “We run Together” vision of TT without the handiathletes and vice versa, while the passengers cannot partake in the wider society or feel the “wind in their hair” without the pilots, who in turn are not piloting anything but an empty rickshaw unless the nursing home residents are present. Furthermore, relatives (of the athletes in TT), and the nursing home staff (in CWA) coordinate practical matters and ensure assist and support for the HAs (TT) and passengers (CWA) before, during and after the activity. Due to this dependency between the groups, WMM will include both the primary (HAs and passengers) and secondary groups (as described in Table 1) in the study. Inclusion of the secondary groups serves three purposes: (1) obtaining further insights into the mechanisms at play for participants of each programme; (2) Validating and furthering the already obtained insights from the primary groups; and (3) Assessing potential health (on top of the already known physical) benefits from helping the primary groups participate in indirect PA. The goal is a deep and comprehensive understanding of what, how and why the two programmes affect the health of all target groups.

## 2. Methods

### 2.1. Study Aims

The overall aim of WMM is to investigate and evaluate how participation in the two pre-existing programmes, TT and CWA, influences QoL as well as physical, social, and mental health among their participants and the persons affiliated with the programmes.

Four interrelated research questions form the base of the evaluation and are aimed at the multiple target groups investigated in the programmes.

Primary objectives:(1)How does being moved by others affect the QoL among handiathletes (disabled people—TT) and passengers (elderly—CWA)?(2)Does indirect PA lead to improved physiological health among handiathletes (disabled people—TT) and passengers (elderly—CWA)?

Secondary objectives:(3)How does physically moving others affect the volunteers’ perceived physical and mental health and their QoL?(4)What does it mean for relatives and nursing staff that handiathletes and nursing home residents are affiliated with TT and CWA, respectively?

This study protocol will present the study design, methods, recruitment process, planned data collections, expected outcomes, and discuss potential design limitations and strengths.

### 2.2. Programme Theory Development

To guide the design of our study, a programme theory, inspired by Funnel et al.’s approach [48], was developed (Figure 2 below). The assumptions presented in the theory are based primarily on desk research [41,42], evidence from literature reviews [24,25], minor pilot studies of the programmes [34,35,36,37,38,39,40,49], theories of change, and stakeholder workshops. The programme theory aims to visualise the assumed causal chain and hypotheses of how any expected change occurs as a result of programme participation and covers expected outcomes for all six (three in each programme) target groups.

## 3. Study Design and Evaluation

The WMM study is a natural experiment, evaluating pre-existing programmes in a real-life setting, with no control over the actors or activities involved [50]. The study uses a quasi-experimental, pretest/posttest (from this point “pre-post”) and multi-methods evaluation approach, examining changes to QoL as the primary outcome measure, following a 12–16 week participation in either the TT or CWA programme. According to Montero and colleagues, a pre/post design is often applied in a natural setting, where controlled trials are not feasible [50]. The WMM study applies the pre/post design; despite the weakness, it may cause (the design is discussed later).

The programmes will be evaluated as two sub-projects. Some instruments will be used across both projects, while others will be exclusively for one or the other. Shared qualitative methods include interviews and participant observations during the 12–16-week programme period. However, each project has target-group-specific aims and scopes, and thus interview and observation guides will be custom fit to each group. Some quantitative methods are shared across the programmes too, such as questionnaires and trip registrations. However, the primary target group in each sub-project will undergo rigorous clinical examination (TT; HAs) and objective functional and cognitive appraisal (CWA; *passengers*). In the following sections, outcome measures and methods shared between the sub-projects, and a power calculation to determine the effect size of the study population will be presented. These will be followed by a description of the recruitment process and our data collection procedures for each individual programme.

### 3.1. Shared Outcomes and Outcome Measures

Our primary outcome measure in all groups is QoL. This will be measured using the Cantril ladder [51]. QoL will be assessed among both primary target groups (HA and passengers) at baseline and at 12–16-week follow-ups, and the secondary target groups during the 12–16 weeks. The Cantril ladder is a generic, 11-point scale where the top (10) illustrates the best possible life while the bottom (0) represents the worst possible life, as perceived by the individual. Higher scores indicate a higher QoL. The ladder is used in many populations and validated in various studies [51,52]. The secondary outcomes are measured before, during and after the intervention period in both programmes. They are based on a causal assumption chain suggesting that improvements to the intermediate outcome measures of the programme theory (Figure 2 above), which includes aspects of physical, social, and mental health, can lead to improvement to QoL.

The participants reported outcomes and repeated measures.

Both primary and secondary target groups will be invited to participate in a web-based questionnaire before and after the programme period (N.B. carried out as interviews with the passengers of CWA). Cross-sectional data will be collected among relatives and nursing staff during the programme period.

The questionnaire will include aspects of physical (e.g., better sleep, metabolic change, absence of pain), social (e.g., reduced loneliness, increased social and emotional support and network), and mental health (e.g., increased well-being, self-efficacy, self-worth). Moreover, sociodemographic characteristics such as age, gender, civil status, (former) occupation, height, weight, and (highest) level of education will be collected. The questionnaires will primarily be constructed from existing, validated scales and items. For instance, items from The Danish National Health Survey [53] (self-perceived health, sleep quality, contact with others), will be used verbatim. Furthermore, the Three-Item Loneliness Scale [54], self-worth [55], self-efficacy [56,57] and, mental well-being scales, will be used [58,59,60]. The questionnaires will be custom fit to each target group to ensure that the included items and questions are relevant and make sense for each specific group. No items will be removed from validated scales, though.

### 3.2. Sample Size and Power

Using STATA, we calculated the power of a two-sided test with conventional levels of statistical power set to 0.8 and level of significance set to 0.05. Focusing on QoL as the primary outcome measured with the Cantril Ladder of Life Scale, we calculated the number of participants needed to achieve a public-health relevant, realistic and detectable effect size for this QoL in the study. This calculation was based on data from a Danish PhD-study [55]. Calculations showed a minimum of 19 primary target group participants in each programme (TT and CWA) were needed. However, due to the lack of literature for our target groups, difficulties were met according to the power calculation. We are not familiar with the actual power of the setup prior to the data collection. A more significant sample is required to obtain and compensate for the probable lack of power. Hence, we aim to include as many participants (primary and secondary target groups) as possible. As part of our analysis, an actual, informed power sample and effect size will be calculated.

### 3.3. Qualitative Approach

To obtain a more profound and nuanced understanding of the target groups and the context, we will be applying qualitative methods. We expect our qualitative data to support our understanding of the mechanisms at play, which will, in turn, enhance our understanding of the hows, whys and whats behind the quantitative outcomes. As illustrated in the programme theory, our assumptions point out essential change processes. These change processes will form the basis of the observations and interview guides, framing a conceptual basis for the investigation. The evaluation will unfold “the black box” illustrated as the programme theory and it seeks to understand and condense what the participants (primary and secondary) experience, feel and perceive when they are a part of the programmes. Furthermore, the relationship and interrelated dependencies between all target groups will be investigated.

As briefly described above, individual and focus group interviews as well as participant observations will be used in both subprojects during the programme period. Despite the methods and outcome measures being identical between the subprojects, programme and target-group-specific interview and observations guides will be developed to ensure relevancy. The qualitative investigation will include:(1)Individual and/or focus group interviews with representatives from all target groups, both primary and secondary, from both programmes.(2)Participant observations during both TT and CWA activities.

We intend for some interviews to be conducted immediately following participant observations, as this will result in a shared experience for the interviews to revolve around. We plan to conduct 3–4 visits at each programme allocated into three periods: first, mid, late in the study period. The researchers will take part in the activity as participating observers. Both context-specific (i.e., immediately after the activity) and non-context specific (e.g., at an arbitrary time during the period where the participants have not completed a programme activity) interviews will be conducted, as we assume participation in the activity may immediately influence the informant’s view of the activity. Interviews will be recorded and transcribed verbatim. Following observations, notes will be typed out and future focus points will be added to the interview and observation guide. Field notes and transcribed material will undergo thematic analysis according to Castleberry and Nolen’s five steps: compiling, disassembling, reassembling, interpreting and concluding [61].

### 3.4. Team Twin—Recruitment and Data Collection

#### 3.4.1. Recruitment Process

Recruitment began in March 2021 and will conclude in September 2021. The establishment of two or three new clubs is expected during this time period. TT stakeholders will be responsible for making the first contact with potential participants. Following this, the project group will oversee further communication and formal recruitment.

#### 3.4.2. Handiathletes

We seek to recruit 19–25 HAs who have not previously participated in TT, or who have had their participation involuntarily paused due to the COVID-19 pandemic. Participants must meet two criteria to be eligible for participation:(1)The HA must be affiliated with a TT local club.(2)The HA must be competent in legal matters (i.e., manage own life).

Newly enrolled athletes from a recently established club will have the highest recruitment priority, followed by new athletes enrolled in an existing club, while experienced athletes from existing clubs, who have had a break of a minimum of five months due to the COVID-19 pandemic, have the lowest priority. The enrolled sample (N = 19–25) will participate in a clinical examination, interviews, and participant observations during TT activities. Participant observations will only be carried out in clubs where one or more of the study participants are affiliated. Moreover, all other HAs (approx. N = 150), who are not participating in the clinical examination, will be invited to participate in a web-based questionnaire.

#### 3.4.3. Runners

We seek to recruit 10–16 runners affiliated with the same clubs our HA are part of. Runners will be invited to participate in 4–6 focus group interviews both during and after the 12–16-week programme period. Furthermore, all active runners across all clubs will be invited to participate in a web-based questionnaire.

#### 3.4.4. Relatives

Lastly, for TT, we seek to recruit 8–10 relatives of the HAs enrolled in the study. These will be invited to participate in 2–3 focus group interviews. Moreover, relatives of the total population of 150 HAs enrolled in TT will be invited to participate in a web-based questionnaire during the intervention period.

### 3.5. Quantitative Approach—Material and Instruments

In this section, we will explain the material and instruments used to conduct a quantitative investigation of the Team Twin sub-project.

#### 3.5.1. Clinical Examination

The participating HAs (*n* = 19–25), who are enrolled in the project, are invited to three days of clinical examination at the Centre for Physical Activity Research (CFAS). We will conduct two-time pre measures pre-programme enrollment, and one measure post-programme period, after 12–16 weeks. Before all three visits, participants are asked to follow a guideline to avoid interferences and to uniformise the visits between participants. The following guidelines will apply prior to the test day:Fasting must be initiated at least 8 h prior to testing.No exercise 36 h prior to testing.No caffeine 24 h prior to testing.No alcohol 48 h prior to testing.No smoking 8 h prior to testing.No Antacida, NSAIDs, paracetamol, or Proton Pump Inhibitors PPIs 24 h prior to testing.

#### 3.5.2. Medical Examination, Blood Samples and Blood Pressure

At visit 1, all participants will go through a medical examination (stethoscopy of lungs and heart) and an anamnesis (including a clarification of the participants’ chronic diseases). The medical examination will be performed as a standard procedure. At all three visits, blood samples will be conducted by standard procedure (25 mL) and will be analysed for HbA1c, fasting glucose, fasting C-peptide, insulin, cholesterol, triglyceride, haematology and electrolytes at Department of Clinical Biochemistry, section 3011, Rigshospitalet. We expect 95% of participants to be examined between 08.00 a.m. and 01.00 p.m. Blood pressure will be monitored as “office blood pressure” where the participant will sit or lay alone with minimal disturbance (e.g., no talking, music, reading etc.). Office blood pressure will be conducted three times with a 5-min interval, and a mean will be calculated. A calibrated Microlife BP A3 Plus blood pressure monitor (Microlife AG Swiss Corporation Espenstrasse 139, CH-9443 Widnau/Switzerland) is used.

#### 3.5.3. Oral Glucose Tolerance Test

To evaluate glucose metabolism, a standard 120-min oral glucose tolerance test (OGTT) will be performed at pretest 1 and 2 and posttest 3. Blood samples (79 mL) are drawn six times 0, 15, 30, 60, 90 and 120 min after intake of 83 g dextrose diluted in 293 mL water equalling 75 g glucose in a total amount of 300 mL. Participants are encouraged to ingest the drink in less than five minutes. Blood samples will be collected and analysed for insulin, c-peptide, and glucose. The participants are asked to stay in the lab for the whole test and only visit the toilet if necessary.

#### 3.5.4. Body Anthropometrics

A dual X-ray absorptiometry scan will be performed at all three visits. The scan will assess the body composition, including total lean and fat mass, bone mass density and fat percentage (tissue). The participants will be fasting and are asked to empty their bladder before the scan. A Prodigy Advance, GE Medical Systems—Lunar, Madison, WI, USA, will be used.

#### 3.5.5. Bio Tracking PA and Sleep Pattern

The HAs will receive a Garmin VÍVOSMART^®^ 4 watch to evaluate programme and leisure activities, everyday activities, pulse, and sleep patterns. A comprehensive “how-to guide” and a demonstration of the watches’ functions and how to wear them correctly will be demonstrated for participants and their relatives/carers. Participants are asked to monitor their activities in two subcategories: (1) TT-programme training activities and (2) “others”. The other category will monitor every other leisure-time activity (e.g., physical therapy, advanced biomechanical rehabilitation, swimming, yoga, Equine-assisted therapy, home training etc.). The watch will track sleep patterns, including the total amount of hours of sleep, sleep levels (Rapid Eye Movement, light, deep, awake), and movement during sleep. Runners will be instructed and trained in the use of the technology, allowing them to assist HAs in equipping the watch prior to training sessions and during races in the study period. The participants are asked to wear the wearable as much as possible during the interventions period. Reminders will be sent to the participants if they forget to upload data. Moreover, telephone and e-mail support will be available during the study period to solve technical errors. Data entry and management regarding the clinical trials conducted at CFAS will be managed at the web-based Clinical Trial Management System (Easy Trial Aps). Easy Trial is an approved management system by the Danish Data Protection Board. All data in paper form (e.g., blood, screen results, body composition) on each participant will be stored in a locked closet at CFAS, Rigshospitalet, Denmark. Information collected in paper form will be entered into the Easytrial Management System twice to prevent disagreements or typing errors when typing manually. All paper material will be scanned and saved at a secure server after the end study. After the study period, all data in paper form will be destroyed (except the informed consent formulas, which will be scanned and saved). All participants have a unique ID number to enable pseudonymised data.

#### 3.5.6. Trip Registration—TT Activities

After each TT activity, one volunteer runner, who has participated in the activity, fulfils a trip registration form. The runner scans a QR-code with a smartphone and the persons a linked to the online trip registration form. The purpose of the trip registration is mainly to notify the HA which has participated in the activity programme. Thus, to register that each participant has participated in the minimum of the eight required programme activities. Further, the runner will be asked questions about the context of the activity (i.e., duration, weather, length, environment, total group size etc.). The runner will lastly give a 0–10 score of the overall activity that day. These data will be used in the explorative analyses and will be used to generate hypotheses of how the context may affect the experience of the programme activity (e.g., if group size, length or the weather plays an active role in a good or bad experience of the programme).

#### 3.5.7. Outcomes

Outcomes and measures in the Team Twin programme evaluation

Table 2 summarises the quantitative outcomes (primary, secondary and explorative) for the TT programme. It displays the points in time of data collection as well as instruments used to cover specific areas of our programme theory. A study overview of the study activities and data collection is presented in Figure 3.

Outcomes of the WMM study—Team Twin

**Table 2 ijerph-18-10008-t002:** Outcomes and measures in the Team Twin evaluation.

Measurement (Outcome)	What (Operational)	How (Instrument)	When (Timing of Collection)	Who (Data Source)
PRIMARY OUTCOME				
Quality of life (QOL)	Cantril Ladder of Life Scale [62]	Web and interview-based questionnaires	Pre, post	HA, *Runners*, *Relatives*
SECONDARY OUTCOMES				
Autonomy	The perceived feeling of being in control over ones own life	Web-based questionnaire	Pre, post	HA
Sleep	Sleep quality and sleep quantity	Web-based questionnaire Bio tracking (HA) SMS-Survey (HA)	Baseline, during (HA), follow-up	HA, *Runners*
Well-being	WHO-five Well-being Index [59]	Web-based questionnaire	Pre, post	HA, *Runners*
Loneliness	A perceived feeling of loneliness and lack of network and support [54,63]	Web-based questionnaire	Pre, post	HA, *Runners*
EXPLORATIVE OUTCOMES				
Self-perceived health	Subjectively perceived Health [64]	Web-based questionnaire	Pre, post	HA, *Runners*
Perceived pain	Mental and physical pain/discomfort [64]	Web-based questionnaire	Pre, post	HA
Self-perceived Physical performance	Subjectively perceived Physical performance [65]	Web-based questionnaire	Pre, post	*Runners*
Epileptic seizures	Reduced epileptic seizures (adjusted version [66])	Paper-based questionnaire	Pre, post	HA
Self-efficacy	General self-efficacy [56,67]	Web-based questionnaire	Pre, post	HA, *Runners*
Self-worth	Perceived feeling of acceptance [55]	Web-based questionnaire	Pre, post	HA, *Runners*
Social/emotional support and network	Contact and support with friends, family and others. The perceived feeling of being valued, respected and accepted by others [64]	Web-based questionnaire	Pre, post	HA
UNINTENDED SIDE EFFECTS				
Fatigue	The perceived feeling of fatigue related to voluntariness or programme activity	Web-based questionnaire	Post	HA, *Runners*
Anxiety	The perceived feeling of anxiety trigged by the programme activity	Web-based questionnaire	Post	HA, *Runners*
Injuries	Amount of injuries by participation	Web-based questionnaire	Post	HA, *Runners*
**Objective clinical data (only measures for HA TT)**				
Body anthropometrics	- Bodyweight - Body mass index - Whole-body lean body mass - Whole-body fat mass - Whole-body bone mineral density	Dual X-ray absorptiometry	Pre1, Pre2 + Post	HA
Clinical blood samples	Blood glucose control: - HbA1c - Fasting glucose - Fasting C-peptide and insulin	Standard clinical procedure	Pre1, Pre2 + Post	HA
	Blood lipids: - Total cholesterol - Triglyceride			
OGTT	- Glucose - C-peptide - Insulin	Standard OGTT-procedure	Pre1, Pre2 + Post	HA
Fitness level	- Blood volume	Estimated from haematocrit level	Pre1, Pre2 + Post	HA
Office blood pressure	- Resting systolic and - diastolic blood - pressure (pulse)	Monitored by a standard procedure for office blood pressure	Pre1, Pre2 + Post	HA
**Trip registration**				
Objective observational data from every training/trip	- Date (DD-MM-YYYY) - Duration (HH:MM) - Length (KM) - Participants (first name) - Environment of activity (e.g., park, forest, lake, urban) - Weather - Total number of participants (*runners* and HA) - Destination(s) - Overall satisfaction	Online trip registration (QR-code directing users to a short web-based questionnaire is scanned immediately after the TT)	During the programmes and after finishing a programme activity.	Context related data completed by a *runner* after each activity TT: *Runners* fulfil the online form

Table 2 summarises the quantitative outcomes (primary, secondary and explorative) for the TT programme. It displays the points in time of data collection as well as instruments used to cover specific areas of the programme theory. Abbreviations: Pre1 = Pretest 1, Pre 2 = Pretest 2 Post = Posttest. TT= Team Twin Programme, HA = Handiathletes. HbA1c = Glycated haemoglobin. OGGT = Oral Glucose Tolerance Test.

**Figure 3 ijerph-18-10008-f003:**
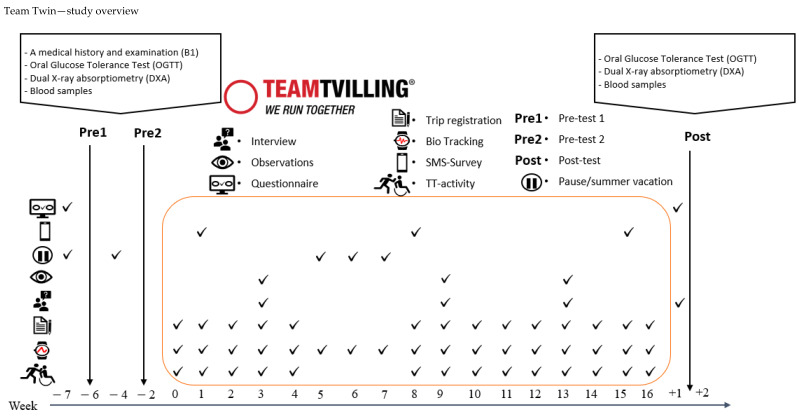
Overview of activities taking place during the TT subproject of WMM.

### 3.6. Cycling Without Age—Recruitment and Data Collection

#### 3.6.1. Recruitment Process

We seek to recruit a total of 40–50 passengers from 6–8 different nursing homes scattered geographically across Denmark. The nursing homes will be selected and recruited in collaboration between the WMM research group and the staff of the CWA association. Recruitment began in January 2021 and will conclude in September 2021. The staff of CWA establishes the first contact to CWA nursing homes. If a nursing home reports interest in participation, contact information is distributed to the project team, enabling the WMM group to formally invite the nursing homes to participate in the project. The nursing staff at the invited nursing homes facilitate the recruitment of residents.

#### 3.6.2. Passengers

The enrolled sample (N = 40–50) will participate in a baseline and follow-up test battery consisting of a questionnaire (conducted as an interview to enable participation of those weak of sight or unable to read) and several physical tests for balance, gait-function and grip-strength. Furthermore, individual interviews, as well as participant observations, will be undertaken with some residents. We seek to enrol passengers who have not previously participated in the CWA activity.

We seek to recruit 14–20 of both pilots and nursing staff to participate in focus group interviews (N = 4–6). These will be recruited from the same nursing homes as the passengers. Participant observations with some pilots will be undertaken. Furthermore, all pilots registered with CWA will be invited to participate in a web-based questionnaire (potentially N = 3000, equals to the registered pilots in the Danish CWA system).

### 3.7. Quantitative Approach—Material and Instruments

This section will explain the characteristics of the material and instruments used to conduct a quantitative investigation of the Cycling Without Age subproject.

#### 3.7.1. Objective Functional and Cognitive Appraisal

A test battery will be conducted pre and post on-site at the participants’ residence to evaluate a change in cognitive and physical function among nursing home residents.

Cognitive function Brief Assessment of Impaired Cognition Questionnaire (BASIC-Q).

A simple and newly invented tool for practitioners and healthcare professionals was selected to assess a possible change in the cognitive function level of the nursing home residents: the BASIC Q. The tool is developed by the Danish Dementia Research Centre and is an effective instrument for detecting mild cognitive impairments. The BASIC-Q is easy to learn and use for practitioners and can be administered in under five minutes ½. The BASIC Q consists of three components (1) self-report, (2) orientation and (3) informant report (e.g., from a relative), with a score from 1–20, where lower scores indicate mild cognitive impairments. However, in situations where informant reports cannot be obtained, a pro-rated score (from 0 to 14) can be calculated [68]. The WMM study applies the pro-rated BASIC-Q score due to practical circumstances. The self-reported component consists of three questions about the persons subjectively estimated memory function [68]. Response options are “No” (2 points), “To some extent” (1 point) and “To a great extent” (0 points). The orientation components consist of three questions regarding time (current year, current month, current day of the week) and one question regarding the person’s age—not their birth year. A score of 2 is obtained for a correct answer, a score of 0 is obtained for a wrong answer [68].

#### 3.7.2. Physical Function and Mobility

To evaluate physical performance and a possible change from pre to post, various physical tests were selected. Our test battery includes (1) the Short Physical Performance Battery [69], (2) the 4 m gait speed test [70,71,72] and (3) the 6-min walking test (6MWT) [70,73,74]. The battery is performed on-site, where the nursing home resident lives, following recommended guidelines and standard procedures for each test [75,76]. All the tests are highly recommended in research and used among elderlies to detect functional capacity and sarcopenia symptoms. The chosen tests have shown great validity and reliability [77].

#### 3.7.3. Muscle Strength

Muscle strength will be evaluated with (1) the chair and stand test (which is incorporated in SPPB) and (2) handgrip strength. Grip strength will be performed with the elbow close to the body, the elbow bent to 90°, with the individual either standing up or sitting down—depending on preference. Studies have suggested that standing up and fully extending the arm can execute greater power [78,79]; however, this was made optional due to the frailty of the target group. The grip-strength measure is performed three times with a 1-min interval between each attempt. The attempt with the highest score (measured in kilograms) will count as the final.

#### 3.7.4. Trip Registration for CWA Programme

Following each CWA trip, where a resident who has taken part in the functional and cognitive appraisal participates, we intend for the pilot or the nursing staff to complete a trip registration. A guide and a QR-code directing the person to a short web-based registration form is attached to the rickshaw bikes of the nursing homes enrolled in the study. The purpose is to track the amount of times study participants go on (at least 6) CWA trips during the 12–16-week study period. The form includes questions related to the specific context (e.g., weather, length, time duration etc.) of the activity. Prior to the CWA trip, each passenger and the pilot themselves are asked to score their “mood”. Verbatim the passengers are asked by the pilots, “How is your mood at this very moment?”. The score will be noted on a smiley-scale rating from 1–5 (sad smiley-face to happy smiley-face). Immediately after the end of the trip, the same procedure is repeated. The data of the context and the trips’ immediate effect on mood will be applied to the explorative analyses and contribute to the development of new hypotheses regarding the influence of context on the programme outcomes.

#### 3.7.5. Outcomes

Outcomes and measures in the Cycling Without Age evaluation

Table 3 summarises the quantitative outcomes (primary, secondary and explorative) for the CWA programme. It displays the points in time of data collection as well as instruments used to cover specific areas of our programme theory. A study overview of the study activities and data collection is presented in Figure 4.

Outcomes of the WMM study—Cycling Without Age

**Table 3 ijerph-18-10008-t003:** Outcome and measures in the Cycling Without Age evaluation.

Measurement (Outcome)	What (Operational)	How (Instrument)	When (Timing of Collection)	Who in CWA (Data Source)
PRIMARY OUTCOME				
Quality of life (QoL)	Cantril Ladder of Life Scale [62]	Web- and interview-based questionnaire	Pre, post	*Passengers*, Pilots, Nursing staff
SECONDARY OUTCOMES				
Autonomy	The perceived feeling of being in control over ones own life	Web-based questionnaire	Pre, post	*Passengers*
Sleep	Sleep quality and sleep quantity	Web-based questionnaire	Pre, post	*Passengers*, Pilots,
Well-being	CUA: Warwick-Edinburgh Mental Well-Being Scale (S)WEMWBS) [58,60]	Web-based questionnaire	Pre, post	*Passengers*, Pilots
Loneliness	A perceived feeling of loneliness and lack of network and support [54,63]	Web-based questionnaire	Pre, post	*Passengers*, Pilots
EXPLORATIVE OUTCOMES				
Self-perceived health	Subjectively perceived Health [64]	Web-based questionnaire	Pre, post	*Passengers*, *Pilots*,
Perceived pain	Mental and physical pain/discomfort [64]	Web-based questionnaire	Pre, post	*Passengers*, *pilots*
Self-perceived Physical performance	Subjectively perceived Physical performance [65]	Web-based questionnaire	Pre, post	*Passengers*, *pilots*
Self-efficacy	General self-efficacy [56,67]	Web-based questionnaire	Pre, post	*Passengers*, *pilots*
Self-worth	Perceived feeling of Acceptance [55]	Web-based questionnaire	Pre, post	*Passengers*, *pilots*
Autonomy	The perceived feeling of being in control over ones own life	Web-based questionnaire	Pre, post	*Passengers*, *pilots*
Social/emotional support and network	Contact and support with friends, family and others. The perceived feeling of being valued, respected and accepted by others [64]	Web-based questionnaire	Pre, post	*Passengers*, *pilots*
*UNINTENDED SIDE EFFECTS*				
Fatigue	The perceived feeling of fatigue related to voluntariness or programme activity	Web-based questionnaire	Post	*Passengers*, *pilots*
Anxiety	The perceived feeling of anxiety trigged by the programme activity	Web-based questionnaire	Post	*Passengers*, *pilots*
Injuries	Amount of injuries by participation	Web-based questionnaire	Post	*Passengers*, *pilots*
**Objective functional and cognitive level (only for CWA; *Passengers*)**				
Cognitive function	Brief Assessment of Impaired Cognition Questionnaire *(BASIC-Q**)* [68,80]	Interview-based case-finding survey	Pre, post	*Passengers*
Physical function and mobility	Short Physical Performance Battery (SPPB) [69,71]	Functional and physical test	Pre, post	*Passengers*
Muscle Strength	Grip Strength [81]	Physical test	Pre, post	*Passengers*
Gait endurance	6-Min Walk Test [73,75]	Functional test	Pre, post	*Passengers*
**Trip registration for CWA**				
Objective observational data from every training/trip	- Date (DD-MM-YYYY) - Duration (HH:MM) - Length (KM) - Participants (first name) - Environment of activity (e.g., park, forest, lake, urban) - Weather - Social activities (eat/drink) - Destination(s) - Mood - Overall satisfaction	Online trip registration (QR-code directing users to a short web-based questionnaire is scanned immediately after the CWA activity)	During the programme After finishing a CWA activity.	Context related data completed by one person after each activity *Passengers* *pilots*

Table 3 summarises the quantitative outcomes (primary, secondary and explorative) for the CWA programme. It displays the points in time of data collection as well as instruments used to cover specific areas of our programme theory.

**Figure 4 ijerph-18-10008-f004:**
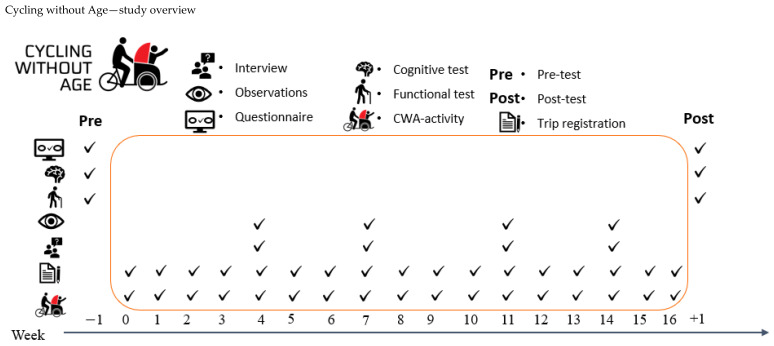
Overview of activities taking place during the CWA sub-project of WMM.

### 3.8. Planned Statistical Analyses

Changes from pre to post in the mean score of QoL, measured using Cantril’s Ladder [51,62], will be the primary statistical outcome measure. Secondary outcomes will include sleep, well-being, loneliness and autonomy, as these were identified, during our design phase, to be essential factors for the target groups. For descriptive statistics and to inform participant characteristics, we plan to use paired t-tests and chi-square tests. We furthermore intend to conduct regression analyses, using a difference-in-difference approach to compare newly enrolled participants with more experienced participants in order to investigate any differences in outcomes. Analyses will be performed as per protocol and after the intention-to-treat principle, which, in this case, means that potential dropouts (with incomplete data) are included in the analyses. Per-protocol participants are defined as participants who have participated in a minimum of eight TT or CWA activities in the period from pre to post measurements. The analyses will be adjusted for sociodemographic factors such as sex, age and educational level. Data are analysed using significance levels of p = 0.05 and 95% confidence intervals. Analyses will be completed by Stata Software (StataCorp LP, College Station, TX, USA).

## 4. Discussion

This study protocol presents the design, methods, and evaluation approaches of the When Movement Moves study and its two sub-projects concerning pre-existing programmes, Team Twin and Cycling Without Age. The two programmes have a shared vision: enhancement of QoL in the disabled and elderly, respectively, achieved through (indirect) PA through able-bodied volunteers’ exercise. By evaluating these original programmes, we get the opportunity to see if, how, and why the programmes may promote health among all involved parties, from disabled, able-bodied and relatives in Team Twin, to nursing home residents, bicycle-pilots and nursing home staff in Cycling Without Age.

### 4.1. Strengths

The variety of methods employed to evaluate potential effects in our study stands as a design strength [82]. The methods were chosen due to their ability, in combination, to explore underlying mechanisms in the evaluated programmes, and uncover the “whats”, “hows”, “whys” and the extends to which each programme may affect its participants. The use of objective measures and clinical data to detect potential physiological or metabolic improvements in the primary target groups may act explanatorily for potential self-reported (Cantril ladder) improvements to our main outcome measure, QoL.

Our methods and assessment tools (e.g., Cantril Ladder of Life Scale, WHO-five, Short Physical Performance Battery etc.) are all previously validated for use with our primary target groups and may therefore result in a higher degree of reliability.

The items constructed specifically for WMM (questionnaires, interview and observation guides), as well as the clinical examination setup (for HAs) and the physical and cognitive appraisal setup (for nursing home residents) all underwent pilot-testing prior to the beginning of the official data collection period, with target group members not enrolled in the study [83]. The questionnaires and interview guides were tested among the primary target groups using a cognitive interviewing technique [84] to explore how questions and response options were understood by the subjects. The pilot setup formed the basis of the development and increased our understanding and knowledge of the field of study, allowing us to make informed (re)designs of data collection methods, as recommended in the MRC evaluation literature [83]. For instance, the questionnaire developed for the passengers were meant to be filled in by the passengers themselves, but issues of sight, fine motor skills and comprehension prompted us to approach the questionnaire as in-person interviews, where the researcher punches in the passenger’s answers. WMM has also undergone an iterative process [85], with changes being made to the passenger questionnaire following initial data collection, as it became evident that some questions were too difficult for the residents with higher degrees of cognitive impairments. To accommodate these residents, as they were still able to participate in the physical examination and parts of the questionnaire, a light-version of the questionnaire was created, consisting of the BASIC-Q (more on this shortly) Cantril ladder, the Three-Item Loneliness Scale and some questions related to their experiences in CWA. The first item performed was changed from the questionnaire to the BASIC-Q. The objective items of the BASIC-Q (relating to year, month, day, and age) can yield up to 8 points. If the resident scored below 6 points, they would take part in the light-version of the questionnaire.

In the physiological measurement setup at CFAS, the clinical pilot protocol was a crucial learning tool as well, as the clinical staff was unaccustomed to examining subjects with disabilities. Testing helped staff and researchers find more appropriate and ethical approaches to collect the highest possible level of data, in terms of quality and validity, within the group. The clinical examinations will be performed at the exact location (CFAS, Rigshospitalet, Copenhagen) and by the same staff group at all three data collection points (pre1, pre2 and post). Standard operations procedures (SOP) are used to standardise the clinical procedures.

### 4.2. Limitations

As the golden standard of evidence is generally considered to be an RCT design [86], the design of WMM can be considered a limitation; an RCT design was neither feasible nor appropriate for ethical and practical reasons, and therefore a quasi-experimental pre/post design was chosen [87]. We intended to recruit a comparison group for a matching design with the elderlies in the CWA programme. However, we were not able to complete that strategy in a meaningful way. Most of the nursing homes we contacted for controls were uninterested in participating—often citing a lack of resources, no residents fit for inclusion (see “Cycling Without Age—recruitment and data collection”), or the still ongoing COVID-19 pandemic as partly responsible.

A comparison group was not feasible or appropriate in the TT programme. Hence, a pre/post study with two baseline measures was the best practical solution and alternative to a more robust design. Hence, a weakened validity and reliability of the study.

To compensate for the lack of control groups, WMM will take a longitudinal approach with pre/post measures in both groups, using themselves as controls. Despite the lack of a solid study design, the current design seemed to be the most appropriate for the WMM study.

Recruitment for both programmes is carried out in steps, with staff from each association (TT and CWA) carrying out the initial recruitment. This increases the risk of convenience recruitment and selection bias due to staffs’ (stakeholder’s) desire for their programmes to perform well. There may also be an increased effect of the programmes due to the COVID-19 pandemic, as most activities—not only TT and CWA—were paused for the past year, and as the primary target groups were isolated due to frailty. Our data collection thus coincides with a re-opening of the Danish society, and activities starting anew, meaning that everyday activities, other forms of PA, and social interactions, may occur unregistered by the study, their effects confounding the actual effect of the programmes.

As the programmes function on interrelated dependency, requiring HAs and passengers to participate in TT and CWA activities on a weekly basis for 12–16 weeks, requires collaboration with runners/pilots to ensure trips are taking place and registered according to the trip registration forms. Moreover, we are reliant on relatives and nursing home staff to ensure HAs and passengers can participate. However, due to the nature of natural experiments, a risk of misleading data and participation bias occurs. Therefore, communication of wants and expectations of participants (and their surroundings) will be comprehensive and—when possible—directly from researchers to relevant individuals in both oral and written format. For instance, a comprehensive “how-to guide” and e-mail is sent to the coordinator of the TT local clubs where HAs enrolled in WMM are associated, and whenever researchers visit a club, they will ensure the information is understood.

### 4.3. Main Contribution

Despite the study’s limitations, we consider the WWM study, and the knowledge it will generate, highly valuable and original due to its target groups. We expect the vast amount of explorative data to pave the way for further research while simultaneously providing methodological knowledge to base further research on, involving the disabled or nursing home residents. Our results will generate new knowledge which can be used for more comprehensive studies of the phenomena of “indirect PA in a social context”. Overall, the study results can be used to form the basis for new practical approaches regarding health promotion among severely disabled and age-related mobility impairment in the elderly.

## 5. Conclusions

In conclusion, WMM is a programme evaluation of the existing programmes Team Twin and Cycling Without Age, anchored in local clubs and nursing homes, respectively, all across Denmark. The study examines the effects on disabled handiathletes’ and the nursing home resident passengers’ participation in indirect physical activity in the TT and CWA programmes on QoL. This paper presents the study design, recruitment process, data collection, outcome measures and planned qualitative and quantitative analysis.

The WMM study provides an original and novel approach in the field of health promotion among its oft-overlooked primary target groups—the disabled and nursing home residents. Through a quasi-experimental pre/post single-group design, WMM will provide essential knowledge and insight into how the two existing programmes that involve movement activities for physically disabled and age-related mobility-impaired elderly can contribute to social, mental, and physical health. The use of both qualitative and quantitative data and a holistic perspective of health-related QoL, assessed in both the primary and secondary target groups, provide a comprehensive perspective on how and why the TT and CWA programmes may lead to improvements to participants’ perceived QoL. Our findings may inform national and international guidelines, e.g., health promotion packages and the Danish Health Authority’s recommendations for physical activity. Internationally, the findings may be used as a clause for allocating resources to environments and programmes promoting health-related QoL for the disabled and the elderly.

## Figures and Tables

**Figure 1 ijerph-18-10008-f001:**
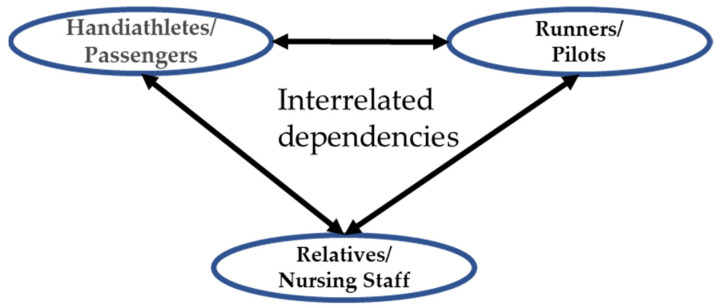
The target groups and their interrelated dependencies in the TT and CWA programmes, respectively.

**Figure 2 ijerph-18-10008-f002:**
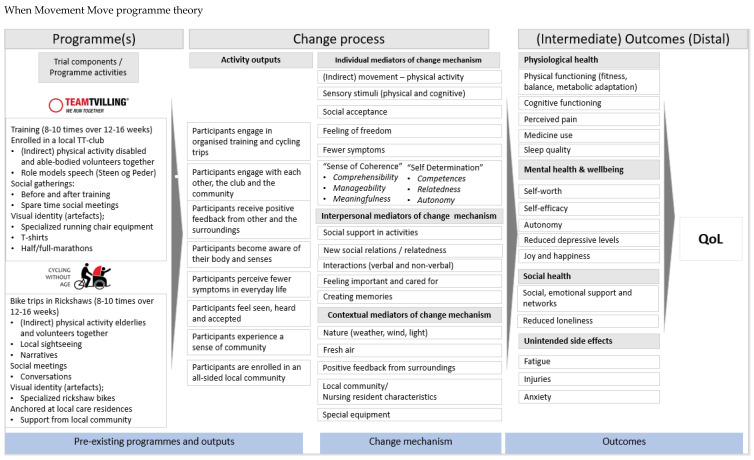
Programme theory overview incorporating both programmes and all target groups. N.B. some elements in the assumed “outcomes” and “change processes” do not count for both programmes and for all target groups.

**Table 1 ijerph-18-10008-t001:** Prioritised target groups of interest according to the two programmes the TT and CWA, respectively.

	Target Group	Primary	Secondary	
Programme				
Team Twin		Handiathletes	Runners	Relatives
Cycling Without Age		Passengers	Pilots	Nursing staff

## Data Availability

No new data were created or analysed in this study protocol. Data collection protocols (e.g., questionnaires, interview guides etc.) are available on request to the corresponding author.

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
