# Peer review of "When Movement Moves: Study Protocol for a Multi-Method Pre/Post Evaluation Study of Two Programmes; the Danish Team Twin and Cycling Without Age"

_ijerph, 2021, doi:10.3390/ijerph181910008_

Round 1

Reviewer 1 Report

The authors present a good research paper. 

  • The relevance of the topic: Good.
  • Introduction: Good.
  • Methodology: Good.
  • Results: Good.
  • Discussion: Good.     

However, ACCEPT AFTER MINOR REVISION. In general, the paper follows an adequate structure and correct scientific support and can be published considering some limitations. The work is interesting in the field of cycling research. However, there are a series of limitations that should be considered.

In the first place, review the existing literature related to the subject, being essential to inquire into the MPDI - IJERPH journal itself, since there are papers related to its manuscript that can help to improve it. Therefore, include those references, if any, especially from the last five years.

Specific comments.

Title: Incorporate in the title that the work supposes a case study.

Abstract. The number of words in the abstract exceeds the limit of words, it is advisable to reduce them, leaving it at most 200 words.

Introduction. This section presents the problem coherently and clearly with the correct support of the scientific literature. However, it is convenient to update the references, since there are different works related to the subject and no mention is made, and it would even be interesting to mention the different existing works related to cycling in different categories. Also, it could be a future study.

Methods.

  • To write the design section, we recommend that you take some of the following methodologists as references.

Ato, M., López-García, J. J., & Benavente, A. (2013). A classification system for research designs in psychology. Anales de Psicología/Annals of Psychology29(3), 1038-1059.

Montero, I., & León, O.G. (2007). A guide for naming research studies in psychology. International Journal of Clinical and Health Psychology, 7(3), 847-862.

  • It is recommended to make a table with all the anthropometric characteristics of the cyclists, and then put the average years, height, weight and Body Mass Index (BMI).

  • Material and instruments. In this section, we will explain the characteristics of the material and instruments used in the investigation.

  • Design a table with the variables used in the study, as well as give a brief description of each variable.

Results. The summary of the study data and table are correct.

Conclusion.  Differentiate the discussion of the main conclusions of the work. To do this, you must create this section. And modify the limitations of the study and locate them in the said section at the end. Also, they must be direct and highlight the main contributions of the study.

References. They should be reviewed and updated according to the publication standards.

Author Response

Deat Editor

Reviewer 2 Report

Dear authors,

thanks for presenting your findings.
Here I present you the points I found that could be ameliorated.
The abstract should focus more on the description of the the two interesting programs TT and CWA and the correlation between the benefits of indirect PA (resulting from the programs implementation ) and the enchancement of QoL. In this way the aims should be sudden clear.
Line 51: what do you mean with generalisability?
The aims are well described in the "Program theory development" paragraph.
Please, could you explain why "-" is often used (e.g.  in-vestigating)? 
Line 91. I suppose "rides" is a little missing. 
Figure 6 should be more centered.
Please, show to readers the statistical analysis data conducted.
It is admirable the quantity and the type of data collected, the your ability to involve so many stakeholders (even though some nursing homes were not interested in participating) as well as the competence in design good researches and the effort to focus on holistic approach, however the topics is slightly new even if the approach could represent a novelty. 

Mange hilsner

Author Response

Dear Editor

Reviewer 3 Report

Proposed keywords are not aligned with PubMed MESH subheadings.

Incorrect use of decimal comma and decimal full stop ("Roughly 3,6% (41.000)").

Authors need to check spelling and grammar (detected overuse of "-" in the whole manuscript).

The authors stated: "The goal is a deep and comprehensive understanding of what, how and why the two programs affect the health of all target groups." and yet in the abstract you mention only the elderly and disabled. This should be aligned accordingly.

Figure 1. should be corrected according to the instruction for the authors.

Does clinical examination will be the same for all participants? How will the authors be assured that the same procedure is done on all participants? On how many locations will be undertaken this examination?

All tables should be corrected according to the instruction for the authors (font type).

A very well-structured manuscript explains really good public health interventions. I am hoping to see soon the final results of this study.

Author Response

Dear Editor 
